# Enhancement of Stopping Power Ratio (SPR) Estimation Accuracy through Image-Domain Dual-Energy Computer Tomography for Pencil Beam Scanning System: A Simulation Study

**DOI:** 10.3390/cancers16020467

**Published:** 2024-01-22

**Authors:** Dong Han, Shuangyue Zhang, Sixia Chen, Hamed Hooshangnejad, Francis Yu, Kai Ding, Haibo Lin

**Affiliations:** 1New York Proton Center, 225 E 126th St., New York, NY 10035, USA; francisyu@nyproton.com (F.Y.); hlin@nyproton.com (H.L.); 2United Imaging Microelectronics, Shanghai 201807, China; shuangyue.zhang@outlook.com; 3Department of Mathematics and Computer Science, College of Arts and Sciences, Adelphi University, One South Avenue, Garden City, NY 11530, USA; schen@adelphi.edu; 4Department of Radiation Oncology, Johns Hopkins University, 401 North Broadway, Suite 1440, Baltimore, MD 21231, USA; hamed@jhu.edu (H.H.); kding1@jhmi.edu (K.D.)

**Keywords:** pencil beam scanning, proton therapy, dual-energy CT, stopping power estimation

## Abstract

**Simple Summary:**

As the technology of the pencil beam scanning system advances, achieving higher accuracy in converting CT Hounsfield to proton-stopping power becomes imperative. Recently, the dual-energy computer tomography (DECT) technique has emerged as a superior technique to single-energy computer tomography in estimating SPR. However, such a technique is not widely adopted clinically. One of the reasons is that the optimization process on the DECT scanners involves a wide range of parameters, one of which is spectral pair. It is generally accepted that the larger separation of the spectra pair could yield a better estimation of SPR. However, it is not validated under a scenario of scanning objects deviating from calibration conditions. In this simulation study, we are examining the performance of variations of spectra pairs on SPR prediction.

**Abstract:**

Our study aims to quantify the impact of spectral separation on achieved theoretical prediction accuracy of proton-stopping power when the volume discrepancy between calibration phantom and scanned object is observed. Such discrepancy can be commonly seen in our CSI pediatric patients. One of the representative image-domain DECT models is employed on a virtual phantom to derive electron density and effective atomic number for a total of 34 ICRU standard human tissues. The spectral pairs used in this study are 90 kVp/140 kVp, without and with 0.1 mm to 0.5 mm additional tin filter. The two DECT images are reconstructed via a conventional filtered back projection algorithm (FBP) on simulated noiseless projection data. The best-predicted accuracy occurs at a spectral pair of 90 kVp/140 kVp with a 0.3 mm tin filter, and the root-mean-squared average error is 0.12% for tissue substitutes. The results reveal that the selected image-domain model is sensitive to spectral pair deviation when there is a discrepancy between calibration and scanning conditions. This study suggests that an optimization process may be needed for clinically available DECT scanners to yield the best proton-stopping power estimation.

## 1. Introduction

Proton therapy’s great advantage of highly conformal dose distribution comes at the cost of high susceptibility to proton range deviations, especially for the pencil beam system. As a result, to account for this proton range uncertainty, safety margins are added to the target (conventionally, 3.5% margin of total range) [1,2,3], and more recently, robust optimization is used to ensure the accurate dose calculation and delivery of prescribed dose to the target. One major source of the proton range uncertainty is the CT-based estimation of material density under the beam path, thus stopping power ratio (SPR). The single-energy CT (SECT) voxel intensity is conventionally translated to stopping power through a stoichiometric calibration process and empirical linear conversion [3].

Dual-energy computer tomography (DECT) technique has shown great potential to improve the prediction accuracy of proton SPR [4,5,6,7,8,9]. DECT-based SPR estimation is built on the idea that the two parameters needed for estimating stopping power, i.e., electron density ($\rho_{e}$) and effective atomic number $Z_{eff}$, can be extracted from CT scans with two distinct CT energies [10,11]. Most of the DECT imaging approaches involve scanning the same object using two different energy spectra. Assuming that the CT numbers of these two scans differ, the electron density ($\rho_{e}$) and effective atomic number $Z_{eff}$ of each voxel in the scanning object can be computed based on equations of photon interaction with the scanned object for each energy. SPR then can be estimated from $\rho_{e}$ and $Z_{eff}$ via the Bethe equation [12,13].

Clinically, the above estimation processes are mainly implemented via image-domain methods [9,14]. These methods comprise the following steps: (i) calibration step that involves the use of two CT spectral on a calibration phantom with known elemental composition to determine the calibration parameters; (ii) scanning of the unknown objects with these two spectra; and (iii) decomposition of the two CT images into maps of two parameters: $\rho_{e}$ and $Z_{eff}$ based on the image-domain method. Although the image-domain method is inferior to the projection-domain method in predicting SP accuracy [15,16], the fast implementation and recent advancements in image reconstruction make the image-domain method still dominant in the clinical application of DECT imaging.

One of the assumptions used by image-domain methods is that scanning conditions of calibration phantoms are identical to those of patients to maintain prediction accuracy. Such scanning conditions include patient size [9,15], region of interest (ROI), variation of the spectra, etc. It is more common to see patient size, especially for pediatric patients, is at least 20% smaller than calibration phantom.

Zhang et al. [15] studied the impact of CT number variations due to size scaling and ROI position changes on the SP prediction accuracy and concluded that since the image-domain methods depend on the quality of reconstructed images, any variations including size scaling that can introduce image artifacts can cause SP estimation errors. However, in the above study, they fail to include the impact of spectra variation of CT scanners on achieved accuracy for image-domain methods.

Ideally, for the DECT imaging technique, the less the overlap of the two energy spectra, the less mutual information from the different spectra can be obtained. This will consequently boost the ability to discriminate the tissues, and material decomposition will be more accurate [14]. Thus, most of the CT scanner vendors choose to use additional filtration, i.e., tin or gold filters, to increase the image contrast or denoise the low-energy images. However, no current DECT scanner type has a free overlap of spectra pair, and for some DECT acquisition techniques, a considerable overlap of the energy spectra is observed. A previous study [14] has found that the spectra gap has a strong impact on the SPR estimation accuracy. However, it remains a question of whether image-domain methods have a similar dependence of energy gap on achieved accuracy.

As many newly built proton centers are acquiring DECT scanners, one of the clinically relevant questions to ask is how we could maximize the efficacy of DECT scanners for serving clinical needs, given the fact that a decent amount of size differences exist between calibration phantom and the patient. For instance, a standard Gammex phantom used for calibration measures 40 cm in diameter. In contrast, the diameter of a pediatric is approximately 20 cm, as per the CDC [17]. The significant size discrepancy may result in a residual beam hardening effect remaining in the CT images, which may compromise the accuracy of SPR predictions [9,18,19].

Thus far, there is no study on the effect of the separation of spectra, maximizing the gain of the achieved accuracy for proton SP prediction through image-domain methods.

In this study, we focus on the effect of selecting spectra pair for DECT imaging on proton-stopping power prediction accuracy. The calibration Gammex phantom is 25% larger in size than the test phantom. The hypothesis is that the maximum spectra separation could make image-domain methods less vulnerable to size change; in other words, the prediction accuracy could be well maintained with less spectral overlap. A series of x-ray tube voltage and tin filtration combinations on a well-acknowledged DECT two-parameter model are investigated. To rule out the intrinsic dependence on spectral change, the sensitivity of DECT model accuracy achieved on spectral variations is also evaluated.

## 2. Materials and Methods

### 2.1. Revisit of the Hünemohr Model

The Alvarez–Macovski photon cross-section model [20] considers the energy-dependent photon linear attenuation coefficient of a known material within the typical photon energy range as a linear combination of virtual photoelectric and Compton scatter contribution:(1)$$\mu \left(E\right)=\rho_{e}\left(a_{ph}\frac{Z_{eff}^{n}}{E^{3}}+b_{com}f_{KN}\left(E\right)\right)$$
where $f_{KN}\left(E\right)$ refers to the Klein–Nishina scatter cross-section. The two proportionality factors, $a_{ph}$ and $b_{com}$, are energy-independent parameters.

The effective atomic number can be determined by the modified Mayneord’s equation:(2)$$Z_{eff}={\left(\frac{\sum_{k}\omega_{k}\left(\frac{Z_{k}}{A_{k}}\right)Z_{k}^{n}}{\sum_{k}\omega_{k}\left(\frac{Z_{k}}{A_{k}}\right)}\right)}^{\frac{1}{n}}$$
where $w_{k}$, $Z_{k}$, and $A_{k}$ are the mass fraction, atomic number, and atomic weight of the $k_{th}$ element in the material, respectively. An n of 3.2 is used throughout this study [21]. Hünemohr et al. [7] proposed the relationship between two spectrum-averaged CT numbers and the material properties. The implementations of Hünemohr et al. [7] can be found in the software *syngo.via* (8.6) by Siemens.
(3)$$\frac{\rho_{e}}{\rho_{e,w}}=\alpha u_{L}+\left(1-\alpha \right)u_{H}$$
(4)$$\frac{Z_{eff}}{Z_{eff,w}}={\left[{\left(\frac{\rho_{e}}{\rho_{e,w}}\right)}^{-1}\left(\beta u_{L}+\left(1-\beta \right)u_{H}\right)\right]}^{\frac{1}{n}}$$
where $\rho_{e,w}$ and $Z_{eff,w}$ are the electron density and effective atomic number of water, respectively; $u_{L}$ and $u_{H}$ are the averaged HU of low- and high-energy CT images, respectively. The two calibration parameters, α and β, depend on the specific dual-energy scanning protocol and can be determined via scanning calibration material other than water. However, the performance would rely on the choice of the calibration material. In this study, we will use Gammex RMI 467 phantom with its original inserts as the calibration phantom.

When applying this method for proton-stopping power estimation via the Bethe Equation, the *I* value is derived from $Z_{eff}$ using the empirical linear relationship that was first introduced by Yang et al. [13]:(5)$$\ln I=a_{Z}Z_{eff}+b_{Z}$$
where the parameters $a_{z}$ and $b_{z}$ are predetermined for different material groups with highly similar compositions, e.g., soft and bony tissues from ICRU tables [22], separately. The *I* value and $\rho_{e}$ can be used in the computation of proton SPR.

### 2.2. Study Design

The geometry of the virtual phantoms was designed based on the Gammex RMI 467 phantom, which consists of a cylindrical solid water background of 330 mm diameter with 17 cylindrical inserts of 30 mm diameter (shown in Figure 1). The standard Gammex tissue substitutes were used as the calibration materials for the Hünemohr [21] method. A total of 34 inserts were simulated in this study; the list of which can be found in ICRU reports [22] and Table 1. The motivation for choosing the Gammax RMI phantom and associated inserts is that they are well-accepted as calibration phantom and easy to access in most proton centers. Also, this phantom is ready to use for our future experimental validation study. The ground truth of phantom inserts is also available from the ICRU 44 report [22].

The spectra used in this study are adapted from Evans et al. [23]. The choice of spectra pair can be found in Figure 2 and Table 2 and Table 3 for the mean energy of each spectrum. We used noiseless projection data for image reconstruction. Since this study is based on the image-domain method, a filtered back-projection image algorithm is used for two CT images reconstruction.

A test phantom that is 25% smaller than the calibration one is also used. It is achieved by scaling the original calibration phantom and inserts proportionally down to 250 mm diameter for the test phantom.

The ground truth of stopping power for ICRU standard tissues can be computed based on the provided material composition data [22]. The root-mean-squared error for 34 tissues is reported under various dual-energy spectral conditions. To quantify the accuracy of prediction SP, the relative error is computed for all pixels belonging to the ROI of each insert. The diameter of ROI is set to 24 mm. An average relative estimation error is calculated for each tissue’s insert ROI. Then root-mean-squared average error (RMSAE) of all 34 tissues can be evaluated.

We also study the effect of spectra mismatch on achieved SP prediction accuracy by keeping the same size of calibration and test phantoms and varying the spectra intentionally for DECT measurements. The pair of spectra used are 90 kVp and 140 kVp for calibration; the adjusted pair of spectra for DECT measurement is achieved by equivalently increasing or decreasing the aluminum filtration. Since the variation could be arbitrary, a reasonably large spectral separation is chosen. In this study, a variation magnitude of 3 mm aluminum thickness is used. The corresponding differences in mean energy are about 1.5 keV and 2 keV for a 3 mm increase in 90 kVp and 140 kVp, respectively. Besides the RMASE of each tissue, the RMS errors (RMSE) of all ROI pixels within each tissue insert ROI are also computed. The hypothesis is that if there is no effect of spectral overlap variation on SP estimation, the accuracy should be maintained from the calibration phantom.

## 3. Results

The relative SP prediction accuracy of tissues under different spectra settings is shown in Figure 3. All ICRU tissues’ accuracy remained within 3%. Figure 3 also shows that $\rho_{e}$ prediction errors for all investigated spectra are within 3%. The tissue of the lung inflated is excluded from the analysis due to its inclusion of a large volume of air. Specifically, if the tin filter is not used to increase the separation, the soft tissues ($Z_{eff}\;$ < 9) can have an SP accuracy within 0.5%, for bony tissues, while the SP accuracy can be as high as 3% for cortical bone ($Z_{eff}\;$ = 13). When a 0.1 mm filter is used to harden the spectrum, the SP accuracy of bony tissues responds to the spectra change promptly, i.e., the accuracy can be improved to around 1%, while for soft tissues, similar accuracy compared to the case of filter-free is maintained. When the spectra are more hardened with thicker tin filters, the most accurate SP estimation occurs around 0.3 mm thickness of the tin filter, not at 0.5 mm. The electron density estimation accuracy dictates the SP accuracy based on the Bethe equation. Although the estimation accuracy of atomic number is not good, among all tin filter thicknesses, the accuracy of electron density at 0.3 mm filter is the best.

The SP RMSAE errors are summarized in Table 4. It shows that with an added tin filter, the SP prediction accuracy is improved compared to the one with no filter used. The theoretical RMSAE for 90/140 kVp and 0.3 mm is 0.12%, and it is about 10 times more accurate.

Although spectra overlap has a role in affecting the SP accuracy when the calibration phantom differs from the test phantom, the DECT model evaluated in this study has minor dependence on the mismatch of spectra if the calibration phantom is identical to the test phantom. Figure 4 shows that with a −3 mm aluminum spectral mismatch between calibration and measurement, i.e., measurement spectra are harder than that of calibration, the SP accuracy of all 34 ICRU tissues is slightly more than 0.6%. If such spectral mismatch is a 3 mm difference, the RMSAE of all tissues for SP prediction is slightly larger than 0.8%, which is shown in Figure 5.

Figure 3 and Figure 4 show that the proton SP accuracy could be up to 2% at maximum between mismatched spectra.

## 4. Discussion

As DECT can improve the accuracy of SPR estimation, here, we examined the robustness of a well-accepted DECT model of proton SP accuracy on two spectral overlaps under a reasonable clinical condition, i.e., a 25% size difference of calibration and test phantoms. Contrary to our prior knowledge that the largest spectra separation can improve the material differentiation and enhance the prediction accuracy of SP, the minimal RMSAE is seen for moderate energy separation, i.e., 90 kVp and 140 kVp with 0.3 mm tin filter. One contributing factor is the vulnerability of the image-domain method to image formation uncertainties. These uncertainties may include non-linear effects of beam-hardening effect, scatter, etc., which may not be accounted for by linear CT image reconstruction algorithm. The calibration procedure can mediate these uncertainties but only if the calibration scan is identical to the test environment. If any variation occurs, for instance, changes in size, as we focused on in our study, it can lead to CT number variations. Image-domain methods are unable to fully compensate for these variations, and thus, it may result in systematic errors in SP prediction. Our study also shows that for a large beam quality variation, in the scenario of identical phantoms, the impact on SP estimation could be as large as 2%. This, in return, can corroborate our hypothesis that if there is a mismatch between the calibration and test phantoms, spectral separation can play a role in the proton SP estimation.

Our study suggests that the choice of spectra for DECT scanners should be carefully validated and justified for maximizing the stopping power accuracy, which can help to mitigate the range uncertainties of protons. Moreover, the achieved SP accuracy is sensitive to phantom size variation for the image-domain method. This is mainly due to the reason that CT number variations can introduce artifacts in images that cannot be accounted for by calibration procedure. Our findings align with the previous finding that any introduced image uncertainties can cause systematic SP errors. However, the achieved high accuracy for the image-domain method can be maintained by optimizing the spectral overlap if the phantom size is deviating from the calibration phantom, in other words, if beam-hardening correction is not fully accounted for by calibration. More in-depth studies are ongoing at our New York proton center.

Almeida et al. [14] compare the $\rho_{e}$ and $Z_{eff}$ prediction performance of twin-beam spectra (120 kVp with tin and gold filters) and dual-source scanner. Their results indicate that the prediction errors for Siemens EDGE twin-beam scanners are noticeably higher than dual-source scanners. For Gammex RMI 467 phantom, EDGE scanners have prediction errors of $\rho_{e}$ up to 15.3%, while for dual-source equipped scanners (FLASH and FORCE), the corresponding errors are within 1.3% and 0.5%, which is similar to our results of added Sn filter. Our study confirms their findings that sufficient spectral separations are needed to achieve a reasonable prediction accuracy. The EDGE twin-beam scanner has the advantage of using a single x-ray tube to realize DECT scans with 0.05 mm Au and 0.6 mm Sn filters on 120 kVp spectrum; however, the consequence of this technique is that poor CT number separation may be produced, leading to the inaccuracy of determining $\rho_{e}$ and $Z_{eff}$ quantitatively. Our study, on the other hand, shows that an optimal spectra separation may exist to achieve maximum accuracy. Thus, to improve the quantitative performance of EDGE twin-beam scanners, it may be advisable to customize the scanner settings to apply the filters to other spectra, which may involve more investigations.

The theoretical quantified impact of each component of uncertainty, except spectral choice, has been summarized and presented in other studies [9,24]; nevertheless, a systematic and comprehensive clinical investigation that includes spectral optimization is warranted. Thus, it would be crucial for any proton centers to conduct uncertainty investigations on their own DECT scanner and determine the strategy to mitigate these uncertainties. Our study can serve as an example of such an investigation, which is one of the novelties of this study. Additionally, it is well-established that the largest spectral separation could yield the most accurate SPR prediction, assuming of size consistency of the calibration and test phantom is maintained [9,14,24]. However, it is commonly found that size deviation can introduce CT number variations. The role of the chosen spectral pair in minimizing these variations remains an open question, demonstrating another novelty of our research. While modification of spectra would greatly involve the engineering effort from the vendor and is often deemed impractical, an alternative approach could involve sequential scans with a combination of optimal choice of two single-energy spectra.

We are aware that our study may have a few limitations. First, among the available DECT models, the one proposed by Hünemohr et al. [21] is implemented. Although our results may be model-dependent, as indicated by Bär et al. [6] this model may yield similar theoretical accuracy compared to other DECT non-linear models. It would be clinically meaningful to conduct inter-model comparisons for the effect of modeling on the SPR prediction accuracy, especially, in the context of size discrepancy and spectra pair optimization. Second, in our study, a pair of sinogram data without noise is employed for image reconstruction and data analysis. Although it is beyond the scope of this study, the noise effect on the performance of estimation accuracy has been discussed elsewhere [9]. However, the combined effects including noise and choice of spectra warrant future study. Third, only a few discretized Sn filter thicknesses are considered to show the proof-of-principle; our conclusions do not change if the continuous variations of Sn thickness are used. And the robustness study of the continuous variations is needed for future study. Last, the FBP reconstruction algorithm is used in this study; as more advanced reconstruction algorithms play a role in the image-domain methods, more investigations are warranted.

## 5. Conclusions

In conclusion, we performed a series of spectral overlap scenarios to assess the vulnerability of the image-domain method of predicting SP accuracy under the clinical condition of phantom variations between calibration and measurement. We found that the 90/140 kVp with 0.3 mm tin filter energy pair can yield the most accurate SP prediction and can account for variations of phantom size. Our study indicates that the choice of energy pair can be optimized to achieve the most quantitative accuracy for predicting SP, which can promote the clinical application of proton therapy in managing devastating diseases, such as pancreatic cancer.

## Figures and Tables

**Figure 1 cancers-16-00467-f001:**
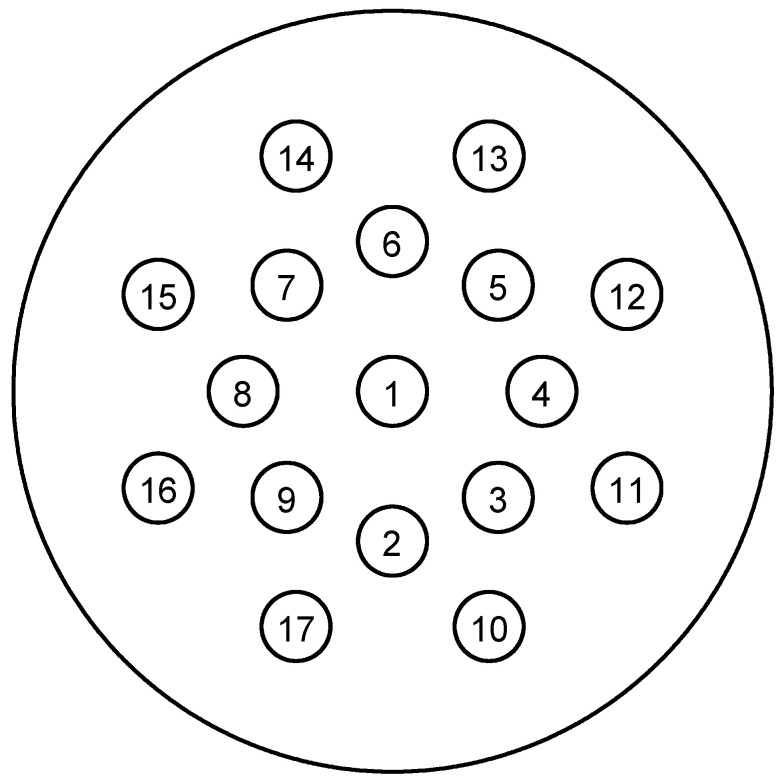
Schematic of virtual Gammex phantom; the insert material can be found in Table 1.

**Figure 2 cancers-16-00467-f002:**
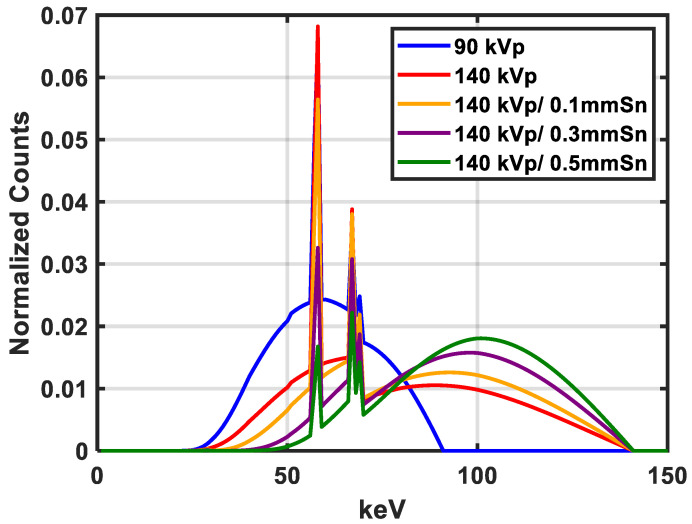
The 90 kVp and 140 kVp spectra were originally measured from the Philip BigBore scanner (Philips, Amsterdam, The Netherlands), and spectra of 140 kVp with Sn filter are simulated with various thicknesses.

**Figure 3 cancers-16-00467-f003:**
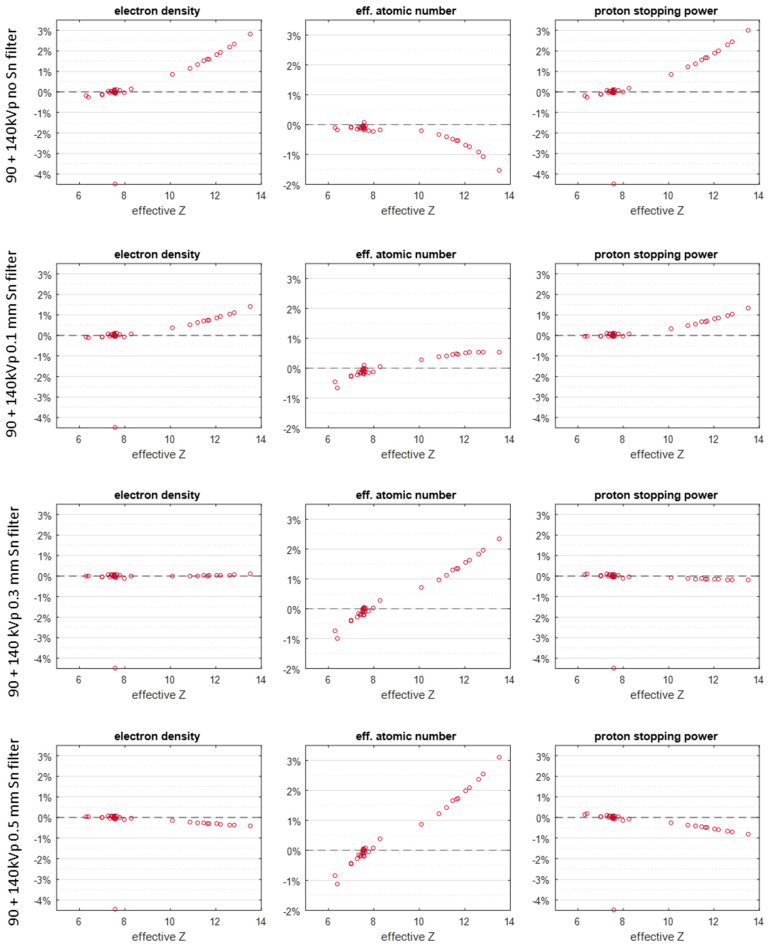
The theoretical relative errors of electron density, effective atomic number, and SP under different spectral conditions: first row no tin filter, second row 0.1 mm tin filter, third row 0.3 mm tin filter, and fourth row 0.5 mm tin filter.

**Figure 4 cancers-16-00467-f004:**
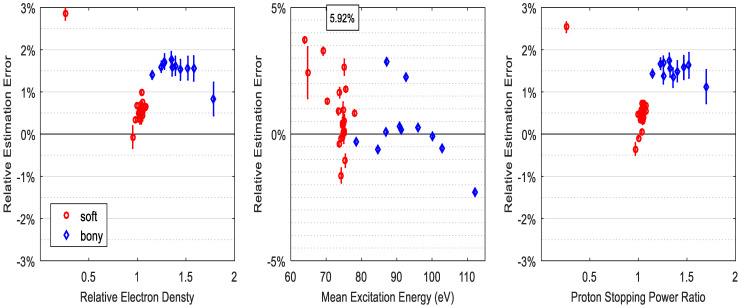
The RMSAE of electron density, mean excitation energy, and SPR prediction accuracy for ICRU standard tissues under the condition of −3 mm spectral mismatch of calibration and DECT measurements. The numbers in the square represent the maximum relative errors.

**Figure 5 cancers-16-00467-f005:**
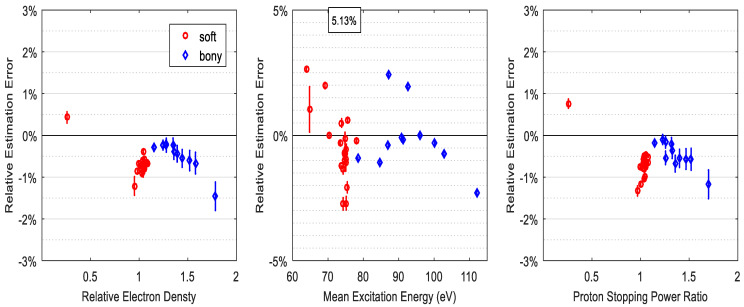
The RMSAE of electron density, mean excitation energy, and SPR prediction accuracy for ICRU standard tissues under the condition of 3 mm spectral mismatch of calibration and DECT measurements. The numbers in the square represent the maximum relative errors.

**Table 1 cancers-16-00467-t001:** The list of insert materials in Figure 1. The original inserts used by Gammex are also included.

Inserts	Gammex	Test Phantom	Test Phantom
1	CT solid water	Adipose	Brain
2	CT solid water	Blood	Cell
3	CB2 50% CaCO_3_	Breast	Lung (deflated)
4	AP6 adipose	Eye lens	GI tract
5	SR2 brain	Heart	Kidney
6	SB3 cortical bone	Liver 2	Lymph
7	BR12 breast	Muscle	Ovary
8	Water	Pancreas	Spleen
9	CB2 30% CaCO_3_	Red marrow	Thyroid
10	CT solid water	Skin	Red marrow
11	IB3 inner bone	Femur	Yellow marrow
12	CT solid water	Mandible	Cartilage
13	LN300 lung	Sacrum	Cortical bone
14	CT solid water	Testis	Cranium
15	LN450 lung	Spongiosa	Humerus
16	B200 mineral bone	Vertebral D6/L3	Ribs (2nd, 6th)
17	LV1 liver	Vertebral C4	Ribs (10th)

**Table 2 cancers-16-00467-t002:** Spectra paired with different ones examined in the study.

Spectra Pair	kVp (Sn Filter Thickness)
1	90/140 (0 mm)
2	90/140 (0.1 mm)
3	90/140 (0.3 mm)
4	90/140 (0.5 mm)

**Table 3 cancers-16-00467-t003:** Mean energy of each spectrum examined in the study.

Spectrum (kVp)	keV
90	56.8
140	71.6
140 (0.1 mm Sn)	83.6
140 (0.3 mm Sn)	91.6
140 (0.5 mm Sn)	97.1

**Table 4 cancers-16-00467-t004:** RMSAE for SP prediction accuracy of ICRU standard tissues.

kVp (Sn Filter Thickness)	RMSAE (%)
90 + 140 kVp (0 mm)	1.35
90 + 140 kVp (0.1 mm)	0.62
90 + 140 kVp (0.3 mm)	0.12
90 + 140 kVp (0.5 mm)	0.4

## Data Availability

The raw data supporting the conclusions of this article will be made available by the authors upon request.

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
