# Peer review of "Enhancement of Stopping Power Ratio (SPR) Estimation Accuracy through Image-Domain Dual-Energy Computer Tomography for Pencil Beam Scanning System: A Simulation Study"

_cancers, 2024, doi:10.3390/cancers16020467_

Round 1

Reviewer 1 Report

Comments and Suggestions for Authors

The manuscript Enhancing of Stopping Power Ratio (SPR) Estimation Through 2 Image-Domain Dual-Energy Computer Tomography for Pencil 3 Beam Scanning System: A Simulation Study by Dong Han et al. reports on a theoretical simulation study aimed at developing a more efficient strategy of spectral pair selection in DECT to improve on the accuracy of stopping power ratio (SPR) prediction needed for the proton therapy.
The design of the study is appropriate and its results are presented more or less clearly.
However, I recommend the authors to present realistic spectra without and with all filters under discussion in a way similar to Fig. 4 in Ref. 21 Relative photon counts vs. Energy. It is also recommended to introduce semiquantitative criteria describing the spectra, e.g., critical or median Energy, width of energy distriution, its asymmetry. To compare distinct spectra in a pair it would be interesting to introduce a semiquantitative criterion for the degree of overlap within the two spectra. I believe, the search for correlations between spectral pair parameters and SPR prediction accuract would be more rational with such a set ofsemiquantitative criteria.

Comments on the Quality of English Language

English is mostly fine.

Author Response

Please see the attached response letter. Thank you!

Reviewer 2 Report

Comments and Suggestions for Authors

The article titled "Enhancing of Stopping Power Ratio (SPR) Estimation Through Image-Domain Dual-Energy Computer Tomography for Pencil Beam Scanning System: A Simulation Study" presents a detailed investigation into the optimization of dual-energy computer tomography (DECT) for improved proton therapy in cancer treatment, particularly regarding the accuracy of stopping power ratio (SPR) estimation. Here are some potential scientific feedback points and suggestions:

  1. Introduction and Background Contextualization: The introduction provides a clear rationale for the study, linking the need for accurate SPR estimation with patient treatment outcomes in proton therapy. However, it would benefit from a more detailed discussion on the current clinical adoption of DECT and the specific challenges faced, to better frame the study's objectives.

  2. Methodological Rigor: The methodologies employed, particularly the simulation of ICRU standard human tissues and the use of different spectral pairs, appear comprehensive. It would be advisable to include a sensitivity analysis to demonstrate the robustness of the results across a range of potential variables.

  3. Data Presentation and Statistical Analysis: The results are presented in a clear manner, with tables and figures effectively summarizing the data. Nonetheless, it would be beneficial to add a statistical significance test to the SP prediction accuracy differences observed with varying tin filter thicknesses.

  4. Limitations and Model Dependence: The discussion addresses potential limitations and acknowledges the dependency on the DECT model used. Future work could expand on the comparison of this model to other DECT models to validate the generalizability of the results.

  5. Clinical Relevance: The discussion could be enhanced by providing more context on how these findings might influence clinical practice. The potential for the model to be used in a clinical setting, including any necessary adjustments or recalibrations, would be useful to explore.

  6. Future Directions: The conclusion points to future experimental validation. It would be constructive to outline a prospective study design or the next steps needed to translate these simulation results into clinical testing.

  7. Technical Aspects and Clarity: The manuscript is well-structured, and the technical language is appropriate for the target audience. However, careful proofreading is recommended to correct minor typographical errors that may distract from the content.

  8. Interdisciplinary Accessibility: While the paper is technically dense, efforts to make it accessible to an interdisciplinary audience, including clinicians who may not be experts in medical physics, could broaden its impact.

In summary, the study is methodologically sound, with clear relevance to improving proton therapy for cancer treatment. Emphasis on broader validation, clinical applicability, and accessible communication would enhance the paper's contribution to the field.

Reviewer 3 Report

Comments and Suggestions for Authors

In this work, the authors present a thorough and well thought out simulation study of uncertainties associated with DECT in proton radiation therapy.  They also designed their work in a way that ensures that clinical validation will be possible in the future.  The mathematical framework in this paper is particularly strong. My only recommendation is that the list of references could be expanded (e.g. include additional proton therapy DECT references.  there have been several publications in recent years, both simulation and measurement based)  

Round 2

Reviewer 1 Report

Comments and Suggestions for Authors

The authors have appropriately addressed my remarks in the revised mauscript that is thus recommended for acceptance in present form.

Reviewer 2 Report

Comments and Suggestions for Authors

Accept in present form

Comments on the Quality of English Language

Accept in present form